# Towards Being Parameter-Efficient: A Stratified Sparsely Activated Transformer with Dynamic Capacity

**Haoran Xu♠, Maha Elbayad♡, Kenton Murray♠, Jean Maillard♡, Vedanuj Goswami♡**

♠Johns Hopkins University, ♡Meta AI

{hxu64,kenton}@jhu.edu
{elbayadm,jeanm,vedanuj}@meta.com

## Abstract

Mixture-of-experts (MoE) models that employ sparse activation have demonstrated effectiveness in significantly increasing the number of parameters while maintaining low computational requirements per token. However, recent studies (Hoffmann et al., 2022; Zuo et al., 2021; Gao et al., 2022) have established that MoE models are inherently *parameter-inefficient* as the improvement in performance diminishes with an increasing number of experts. We hypothesize this parameter inefficiency is a result of all experts having equal capacity, which may not adequately meet the varying complexity requirements of different tokens or tasks. In light of this, we propose Stratified Mixture of Experts (SMoE) models, which feature a stratified structure and can assign dynamic capacity to different tokens. We demonstrate the effectiveness of SMoE on three multilingual machine translation benchmarks, containing 4, 15, and 94 language pairs, respectively. We show that SMoE outperforms multiple state-of-the-art MoE models with the same or fewer parameters.[1]

## 1 Introduction

Scaling up the model and data size has shown tremendous success in enhancing model performance across a large number of NLP tasks (Devlin et al., 2019; Conneau et al., 2020; Kaplan et al., 2020; Brown et al., 2020). Sparsely gated mixture of experts (MoE) (Shazeer et al., 2017; Lepikhin et al., 2021) provides an effective way to greatly scale the model size under the same computational cost and achieves state-of-the-art performances on various tasks including natural language understanding (Fedus et al., 2021), machine translation (NLLB Team et al., 2022), language modeling (Du et al., 2022), etc.

The efficiency comes from sparsely activating a subset of the neural network weights for each incoming sample. However, MoE is reported to be **parameter-inefficient** (Hoffmann et al., 2022; Zuo et al., 2021; Gao et al., 2022) i.e., there are diminishing improvement returns from adding more experts. For example, Switch Transformer (Fedus et al., 2021) only outperforms T5 (Raffel et al., 2020) by an average of 0.7 on the GLUE benchmark (Wang et al., 2018) despite being $35\times$ larger. Similarly, in the translation task, a MoE model with 20 times more parameters only offers an average improvement of 0.3 BLEU on its ablation dataset (MoE-64 vs. 1.3B dense) (NLLB Team et al., 2022).

We hypothesize that this parameter inefficiency stems from the equal capacity assignment, where we particularly define 'capacity' as the **number of parameters used for the incoming token**. For the current MoE models, the capacity of experts are the same used for serving all tokens. However, different tokens may demand varying capacities. For instance, in the context of multilingual machine translation, certain translation directions may necessitate a greater capacity to prevent overfitting, while others only require a smaller capacity. To address this limitation, our hypothesis posits that the dynamic allocation of capacity to tokens results in more efficient utilization of parameters. Thus, we propose Stratified Mixture of Experts (SMoE) models, characterized by a stratified structure, which allows for the dynamic assignment of capacity to incoming tokens.

A high-level comparison of vanilla MoE and SMoE is presented in Figures 1a and 1b. In vanilla MoE, a single routing gate connects to all $E$ experts and sends tokens to the top-$k$ experts. Here, we take $E{=}5$ as an example. In SMoE, the experts are divided into two strata. Each stratum has its own routing gate that connects to all experts in the current stratum as well as all experts in

---

[1]Code is released at https://github.com/fe1ixxu/Stratified_Mixture_of_Experts.

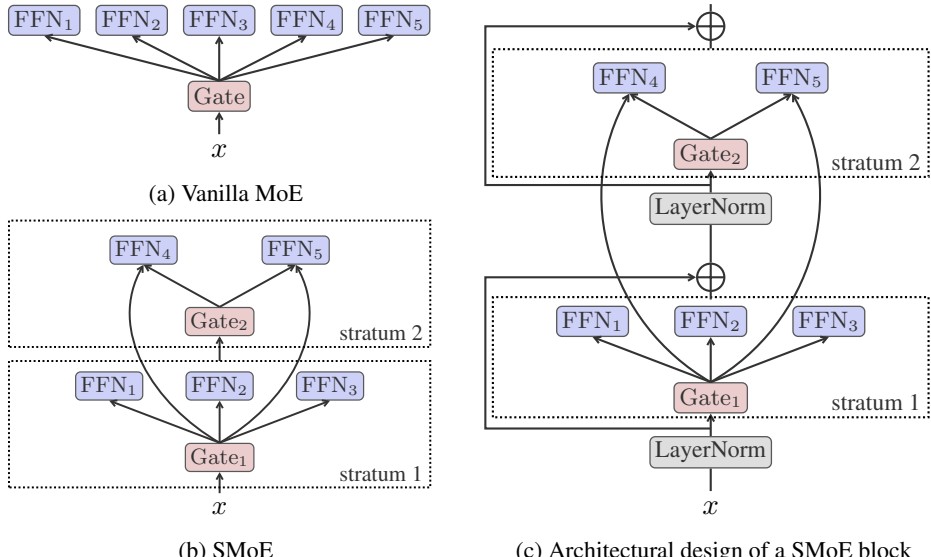

(a) Vanilla MoE

(b) SMoE

(c) Architectural design of a SMoE block

Figure 1: A high-level illustration of the vanilla MoE and SMoE. a) Vanilla MoE: gate is connected to all experts and sends tokens to top-$k$ selection. b) SMoE: Experts are stratified into $L$ strata ($L=2$ in this example). Each stratum has a gate that is connected to all subsequent experts. Tokens can be directly sent to the last stratum to only experience one expert, or be sent to both strata and have more capacity. Hence, the dynamic capacity of a token depends on how many experts it needs to pass. c) A detailed architectural design, where a comprehensive explanation of the design components will be presented in Section 3.

the subsequent strata. If $\text{Gate}_1$ assigns tokens to Expert 4 or 5, the tokens will only need to pass through a single expert (an FFN layer). However, if tokens are sent to experts in the first stratum (Experts 1 to 3), they will need to go through the next stratum as well, meaning that another expert will be assigned by $\text{Gate}_2$ before exiting the SMoE block. This allows SMoE to dynamically assign capacity to different tokens. In addition, a comprehensive illustration of the architectural design of the SMoE model is provided in Figure 1c. A thorough explanation of the design elements will be provided in Section 3. Our main contributions are summarized as follows:

- We introduce the concept of **dynamic capacity** for MoE models and propose a mixture-of-experts model with a stratified structure, namely SMoE, which can automatically assign dynamic capacity to different incoming tokens to make experts become more parameter-efficient.

- We focus on the task of multilingual machine translation (MMT) and show that SMoE substantially outperforms numerous strong baselines with fewer than or the same number of parameters. For instance, we demonstrate that SMoE only needs half the number of

parameters to achieve a performance on-par with a naive MoE (Lepikhin et al., 2021). Furthermore, we carry out an in-depth analysis to probe the factors that impact dynamic capacity assignment, including the language of tokens and the position of the SMoE block within the model's architecture.

## 2 Background and Related Work

Massively multilingual machine translation models have been developed to handle several translation directions simultaneously in a single model (Aharoni et al., 2019). However, the use of shared parameters for different languages often leads to negative transfer and decreased performance (Conneau et al., 2020; Fan et al., 2020). In contrast to dense MMT models, sparsely gated mixture-of-experts (MoE) models, which activate a subset of parameters for each input, have been shown to significantly improve translation performance (Kim et al., 2021; NLLB Team et al., 2022). Shazeer et al. (2017) first demonstrated the benefit of adding MoE layers to scale RNN models for improved translation performance, and Lepikhin et al. (2021) extended this work to transformer architectures (Vaswani et al., 2017). MoE layers in the transformer model replace a single feedforward network (FFN) layer with $E$ FFN layers, denoted

with $\{\text{FFN}_1, \dots, \text{FFN}_E\}$. Each FFN layer is an expert. Given an input token $x$, we have

$$\forall e \in \{1, \dots, E\},$$
$$\text{FFN}_e(x) = W_1^{(e)}\text{ReLU}(W_2^{(e)} \cdot x), \qquad (1)$$

where $W_1^{(e)}$ and $W_2^{(e)}$ are the weights of $\text{FFN}_e$. A trainable routing gate with weights $W_g$ predicts scores for these experts to use for the input $x$, in the form of a routing vector $G \in \mathbb{R}^E$:

$$G = \text{softmax}(W_g \cdot x). \qquad (2)$$

We select the set of top-$K$ experts, denoted with $\mathcal{E} \subset \{1, \cdots, E\}$, and compute the output of the MoE layer as follows:

$$x_{\text{out}} = \sum_{e \in \mathcal{E}} G_e \cdot \text{FFN}_e(x). \qquad (3)$$

MoE models suffer from the notorious load imbalance issue, where the gate weights could collapse and send most tokens to the same expert. As a result, recent research has focused on designing better auxiliary load balancing loss functions to encourage tokens to be evenly distributed across experts, e.g., Lewis et al. (2021) formulated token-to-expert allocation as a linear assignment problem, Roller et al. (2021) modified the feedforward layer to hash to different sets of weights depending on the current token, Zoph et al. (2022) proposed a router z-loss that resolves instability issues, and Zhou et al. (2022) reversely design an expert-to-token allocation algorithm. Other lines of investigation in MoE include regularization techniques such as gating dropout (Liu et al., 2022) and output masking (EOM and FOM) (NLLB Team et al., 2022), as well as novel MoE architectures, such as conditional MoE routing (CMR) that add an extra branch beside MoE layer (NLLB Team et al., 2022), or Pyramid-Residual MoE (Rajbhandari et al., 2022), a hybrid dense and MoE model with more experts in the last layers.

However, all previous work default to equal capacity for all tokens regardless of their language, frequency, or any other property. In the subsequent sections, we present a Stratified Mixture of Experts (SMoE) model that automatically assigns dynamic capacities to different types of tokens.

# 3 Stratified Mixture of Experts

## 3.1 Architectural Design

The guiding design principle for Stratified Mixture of Experts (SMoE) is to assign dynamic capacity to tokens. Given $E$ experts in an MoE block, we partition them into $L$ strata. The $i^{th}$ stratum has a gate $\text{Gate}_i$ which routes tokens to an expert in the current stratum as well as the subsequent ones. This means that tokens can never be sent back to the previous strata. Tokens keep getting routed in the SMoE block until they reach the final stratum. Different tokens will pass through a varying number of experts, resulting in different capacities, according to the assignment of gates. In vanilla MoE, however, the capacity through which every token goes is that of a single FFN layer. The workflow of how a token passes a SMoE block is shown in Figure 1c. For example, some tokens in the $1^{st}$ stratum may be assigned to experts in the $2^{nd}$ stratum while others are sent to the $3^{rd}$ stratum. After several rounds of assignments, tokens finally exit the block after reaching the last stratum.

In SMoE, the successive application of multiple FFN layers to a token can result in training instability. Therefore, following the approach of Vaswani et al. (2017); Wang et al. (2019); Xiong et al. (2020), we incorporate layer normalization (LayerNorm, Ba et al. (2016)) before dispatching tokens to experts and a residual connection after the tokens have passed through the experts. See Figure 1c for an overview of the design of our stratified experts.

Formally, given $T$ tokens in a mini-batch, we denote the $d$-dimensional representation of the $t^{th}$ token in the $i^{th}$ stratum of the current SMoE block with $x_{i,t}$. Let $\mathcal{E}_i$ be the set of experts visible to the current gate (current stratum plus subsequent strata) and let $E_i = |\mathcal{E}_i|$ be its cardinality. Before being dispatched to FFN layers, tokens are firstly normalized with LayerNorm,

$$x'_{i,t} = \text{LayerNorm}(x_{i,t}). \qquad (4)$$

Then, $\text{Gate}_i$ with weights $W_i$ predicts a probability distribution $G_t \in \mathbb{R}^{E_i}$, scoring all visible $E_i$ experts at that stratum:

$$G_t = \text{Gate}_i(x'_{i,t}) = \text{softmax}(W_i \cdot x'_{i,t}). \qquad (5)$$

Following Lepikhin et al. (2021), we dispatch each token to at most $k=2$ experts. If $\mathcal{E}$ is the set of selected top-$k$ experts and the expert with

the highest score is in the $j^{th}$ ($j > i$) layer, $x_{i,t}$ will be assigned to the $j^{th}$ layer and the output computation on the token is:

$$x_{j,t} = \sum_{e \in \mathcal{E}} G_{t,e} \text{FFN}_e(x'_{i,t}), \qquad (6)$$

where $G_{t,e}$ is the gate score for the expert $e$. Finally, We employ a residual connection after the FFN layer:

$$x_{j,t} = x_{j,t} + x_{i,t}. \qquad (7)$$

Tokens gradually pass experts in deeper layers until they finish passing the final layer.

## 3.2 Load Balancing

Similar to Lepikhin et al. (2021), we encourage tokens to be uniformly distributed across all visible experts. Each gate has a loss term to balance the load. For $\text{Gate}_i$, the loss is:

$$\mathcal{L}_i = E_i \sum_{e \in \mathcal{E}_i} f_e p_e, \qquad (8)$$

where $f_e$ is the fraction of tokens dispatched to expert $e$, as their first choice, through top-$k$-gating:

$$f_e = \frac{1}{T} \sum_{t=1}^{T} \mathbb{1}\{\arg\max G_{t,e} = e\}, \qquad (9)$$

and $p_e$ is the average routing probability to that expert over the T tokens in the mini-batch:

$$p_e = \frac{1}{T} \sum_{t=1}^{T} G_{t,e}. \qquad (10)$$

The auxiliary loss for the current SMoE block is computed by taking the average of the loss over all gates within the block.

$$\mathcal{L} = \alpha \cdot \frac{1}{L} \sum_{i=1}^{L} \mathcal{L}_i, \qquad (11)$$

where $\alpha$ is a hyperparameter to control the strength of the load balancing loss. We average the loss over all SMoE blocks in the architecture as the final auxiliary loss appended to the original task loss.

## 4 Experiments

We evaluate the proposed SMoE on a *many-to-many* multilingual neural machine translation task.

## 4.1 Datasets

In this study, we consider three datasets comprising 4, 15, and 94 languages each. The initial two datasets are extracted from the primary bitexts of the NLLB-200 training dataset, and we adopt their resource-level categorizations: *high-resource* ($\geq$ 1M), *very low-resource* ($\leq$ 100K), and low-resource (the remaining).[2] These two datasets are developed and evaluated using the Flores-200 dataset. The third dataset is OPUS-100 (Zhang et al., 2020). We follow Zhang et al. (2020) and divide directions into *high-resource*, *medium-resource*, and *low-resource* categories.

**NLLB M4 dataset** From NLLB, we pick 4 languages from 4 different linguistic families, 2 of which are high-resource and the other 2 are low-resource: Northern Sotho (nso, 526K parallel sentences), Malay (msa, 1M), Tagalog (tgl, 1M), Catalan (cat, 634K), totaling 3.2M training examples.

**NLLB M15 dataset** Taking into account linguistic diversity and larger data size, we expand the M4 dataset to cover a set of diverse 15 languages. M15 covers 6 linguistic families and a balanced number of high-resource, low-resource, and very low-resource languages (each category has 5 languages). We show a detailed listing and information on the M15 dataset in Appendix A.

**OPUS-100** In addition to the datasets derived from NLLB, we also utilize OPUS-100 to examine a scenario involving a larger number of languages. OPUS-100 encompasses a total of 100 languages, which supports 94 development/test language pairs.

**Evaluation** During inference, we use beam search with a beam size of 5 and a length penalty of 1.0. We report BLEU scores (Papineni et al., 2002) for models trained on NLLB dataset and sacrebleu (Post, 2018) with flores200 tokenizer for models trained on OPUS-100.

## 4.2 Baselines

We use five strong baselines to evaluate the effectiveness of SMoE. All baselines are our own implementation following the settings from the original papers. Note that the total number of experts i.e., the full model capacity is kept constant

---

[2]Contrary to NLLB Team et al. (2022), in our study, *very low-resource* is not included in the *low-resource* category but considered as an independent set.

for all models in order to ensure fair comparison (8 experts for M4 and 16 experts for M15).

**Vanilla MoE.** An MoE model with Top-2 gating (Lepikhin et al., 2021).

**Switch Transformer.** An MoE model with Top-1 gating (Fedus et al., 2021). Switch Transformer was introduced to mitigate training instabilities and improve the efficiency of Top-2 MoE models. Note that switch transformer uses fewer FLOPs per token due to the top-1 gating approach.

**MoE + EOM.** A vanilla MoE model regularized with Expert Output Masking (EOM) (NLLB Team et al., 2022). EOM masks the expert output for a random fraction ($p_{eom}$) of outputs. We set $p_{eom} = 0.1$ following the suggestion of NLLB Team et al. (2022)

**Conditional MoE Routing (CMR).** CMR (NLLB Team et al., 2022) augments MoE with a binary gate that sends tokens to one of two branches: (1) a shared FFN layer and (2) an vanilla MoE layer. Note that this method requires extra parameters due to the added shared FFN layer. The CMR budget constraint is set to 0.8.

**Stacking MoE Layers.** A similar model architecture to SMoE where we simply stack multiple MoE layers, e.g., stacking 2 MoE layers with 4 experts each vs. one block of SMoE with 2 strata where each stratum has 4 experts. Unlike SMoE where each expert is surrounded with a residual skip connection and preceded with a LayerNorm, here the stacking is naive without any addition.

### 4.3 Training Details

Following Johnson et al. (2017), we prepend source sentences with a special language token <2xxx> to indicate the target language. We use a data sampling temperature of $T=1$ suggested by NLLB Team et al. (2022) to train on NLLB datasets, and $T=5$ suggested by Zhang et al. (2020) to train on OPUS-100.

The dense model architecture, backbone for all trained models, is a Transformer model (Vaswani et al., 2017) with 12 layers (6 on encoder and 6 on decoder). We use transformer$_{base}$ and $E=8$ experts for M4, and transformer$_{big}$ and $E=16$ experts for M15 and OPUS-100.[3]

---

[3]transformer$_{base}$: FFN dimension of 2048, 8 heads, and embedding dimension of 512; transformer$_{big}$: FFN dimension

In MoE models, every other FFN layer of the encoder and decoder are substituted with an MoE layer. For SMoE, in the $i^{th}$ stratum, we enforce that each expert processes, at most, $2 \times T_i/E_i$ tokens, where $T_i$ is the number of tokens in the mini-batch sent to the layer $i$ and $E_i$ is the number of visible experts. For the other MoE baselines, it is $2 \times T/E$, where T is the number of tokens in the mini-batch and $E$ is the total number of experts. The multiplicative coefficient $\alpha$ for the auxiliary load balance loss is set to 0.01. A vocabulary of size 32k for both M4 and M15 and 64K for OPUS-100 with SentencePiece (Kudo and Richardson, 2018). For a fair comparison, All models are trained for the same number of updates. More details can be found in Appendix B.

### 4.4 SMoE configurations

We use a series of numbers separated by hyphens to describe the SMoE configuration. For instance, SMoE-4-4-8 indicates that all MoE blocks have 3 strata, where the $1^{st}$ stratum has 4 experts, the $2^{nd}$ has 4 and the $3^{rd}$ has 8.

### 4.5 Results

**M4 Results** The results are in Table 1. We consider two configurations for SMoE: SMoE-4-4 and SMoE-2-2-2-2. The better SMoE settings for M4 is SMoE-2-2-2-2. In M4, SMoE-2-2-2-2 outperforms Switch Transformer by +0.9 BLEU on average and vanilla MoE by +0.74 BLEU. Out of 5 MoE baselines, CMR achieves the best performance (+0.4 BLEU over Switch Transformer and +0.3 BLEU over vanilla MoE), however, CMR models have more parameters as well as more FLOPs per token.

It is worth noting that simply stacking MoE layers degenerates the model performance, which indicates the importance and effectiveness of the specific design of SMoE.

**M15 results** We show results in Table 2. We consider a multitude of settings splitting the 16 experts per layer over 2, 3 or 4 strata. The best SMoE settings for the larger M15 dataset is SMoE-4-12. This configuration demonstrated an average improvement of +1.04 BLEU over the Switch Transformer and +0.93 BLEU over the vanilla MoE across the 15 languages evaluated. However, MoE+EOM and CMR only improve vanilla MoE by +0.52 and

---

4096, 16 heads, and embedding dimension 1024.

| Methods (8 experts) | xxx→eng | | | | eng→xxx | | | | Avg. | #Parameters | FLOPs/tok |
|---|---|---|---|---|---|---|---|---|---|---|---|
| | nso | msa | tgl | cat | nso | msa | tgl | cat | | | |
| Dense Model (Vaswani et al., 2017) | 28.87 | 37.01 | 41.76 | 40.35 | 26.68 | 44.30 | 38.20 | 45.50 | 37.83 | 60M | 167M |
| Vanilla MoE (Lepikhin et al., 2021) | 29.45 | 41.26 | 44.23 | 42.91 | 27.59 | 47.12 | 40.10 | 47.66 | 40.04 | 148M | 192M |
| Switch Transformer (Fedus et al., 2021) | 29.69 | 40.35 | 44.50 | 43.19 | 27.73 | 46.44 | 39.88 | 47.26 | 39.88 | 148M | 167M |
| MoE+EOM (NLLB Team et al., 2022) | 30.52 | 40.50 | 44.52 | 43.48 | 27.71 | 46.85 | 40.09 | 47.14 | 40.10 | 148M | 192M |
| MoE+CMR (NLLB Team et al., 2022) | 30.54 | 41.29 | 44.65 | 43.71 | 27.76 | 46.87 | 40.36 | 47.65 | 40.35 | 161M | 217M |
| Stacking 2 4-expert MoE | 28.65 | 39.24 | 42.02 | 41.36 | 27.58 | 46.27 | 40.06 | 47.50 | 39.09 | 148M | 242M |
| SMoE-4-4 | 30.14 | 41.05 | **45.13** | **43.97** | 27.86 | 47.48 | 39.85 | 47.71 | 40.40 | 148M | 217M |
| Stacking 4 2-expert MoE | 25.53 | 35.53 | 38.34 | 37.43 | 24.99 | 41.35 | 36.85 | 42.15 | 35.27 | 148M | 327M |
| SMoE-2-2-2-2 | **30.96** | **41.88** | 44.86 | 43.37 | 27.83 | **47.96** | **40.95** | **48.39** | **40.78** | 148M | 247M |

Table 1: Overall BLEU results on the M4 dataset. The best values are bold and the second-best values are underlined. The number of experts is 8 for all methods. The two SMoE models attain the two best performances across all languages.

| Methods (default is 16 experts) | | eng→xxx | | | | xxx→eng | | | | Avg. | #Parameters | FLOPs/tok |
|---|---|---|---|---|---|---|---|---|---|---|---|---|
| | | all | high | low | very low | all | high | low | very low | | | |
| Dense Model (Vaswani et al., 2017) | | 29.27 | 38.68 | 32.44 | 16.69 | 29.94 | 37.55 | 32.37 | 19.89 | 29.61 | 209M | 506M |
| Vanilla MoE (Lepikhin et al., 2021) | | 32.01 | 40.41 | 34.77 | 20.84 | 32.00 | 39.38 | 34.73 | 21.89 | 32.00 | 963M | 606M |
| Switch Transformer (Fedus et al., 2021) | | 31.67 | 40.44 | 34.72 | 19.87 | 32.09 | 39.85 | 34.75 | 21.68 | 31.89 | 963M | 506M |
| MoE+EOM (NLLB Team et al., 2022) | | 32.02 | 40.63 | 35.19 | 20.25 | 32.81 | 40.42 | 35.33 | 22.69 | 32.41 | 963M | 606M |
| MoE+CMR (NLLB Team et al., 2022) | | 31.83 | 40.22 | 34.81 | 20.46 | 33.02 | 40.41 | 35.42 | 23.21 | 32.42 | 1.01B | 707M |
| 2-layer SMoE | SMoE-4-12 | **33.00** | 41.14 | 35.77 | **22.10** | 32.86 | **40.63** | 35.14 | 22.80 | **32.93** | 963M | 656M |
| | SMoE-12-4 | 32.52 | 41.11 | 35.56 | 20.89 | 32.94 | 40.03 | **35.21** | 23.60 | 32.73 | 963M | 757M |
| | SMoE-8-8 | 32.28 | 41.24 | 35.33 | 20.27 | 32.72 | 40.30 | 34.80 | 23.08 | 32.50 | 963M | 707M |
| 3-layer SMoE | SMoE-4-4-8 | 32.83 | **41.61** | **35.82** | 21.07 | 32.24 | 39.40 | 34.53 | 22.80 | 32.54 | 963M | 724M |
| | SMoE-8-4-4 | 32.24 | 40.99 | 35.12 | 20.61 | **33.06** | 40.36 | 35.12 | **23.71** | 32.65 | 963M | 805M |
| 4-layer SMoE | SMoE-4-4-4-4 | 32.87 | 41.50 | 35.68 | 21.42 | 32.63 | 40.03 | 34.50 | 23.35 | 32.75 | 963M | 825M |
| Vanilla MoE, 32 experts | | 32.98 | 41.05 | 35.69 | 22.19 | 32.90 | 40.79 | 35.49 | 22.42 | 32.94 | 1.77B | 606M |

Table 2: Overall BLEU results on the M15 dataset. The best values are bold and the second-best values are underlined. Unless otherwise mentioned, the number of experts is 16. All SMoE models outperform the baselines. The best setting is SMoE-4-12, which outperforms vanilla MoE by +0.93 BLEU. Vanilla MoE would require to double its parameters to achieve similar performance to SMoE-4-12.

+0.53 BLEU, respectively. Note that if vanilla MoE wants to achieve similar performance to SMoE-4-12 (32.94 vs. 32.93 BLEU on average), it has to increase experts from 16 to 32, which almost doubles the total number of parameters from 963M to 1.77B (last row of Table 1), which means our model is much more parameter-efficient.

SMoE models with more strata, allowing for more depth, do not guarantee better performance. A clear example is SMoE-4-12 and SMoE-4-4-4-4 in M15 (32.93 vs. 32.75 averaged BLEU). However, for 'balanced' SMoE (equal number of experts per stratum), fewer experts per stratum achieves better performance: SMoE-2-2-2-2 outperforms SMoE-4,4 (40.78 vs. 40.40 BLEU) on M4 and SMoE-4-4-4-4 outperforms SMoE-8,8 (32.75 vs. 32.50 BLEU) on M15.

**OPUS-100 Results: A Larger Performance Gap.** Table 3 presents the comprehensive results. Notably, the performance disparity becomes more pronounced we scale our experiments to

94 languages ( OPUS-100 does not support the remaining 5 languages ). We select the two optimal SMoE configurations in M15: SMoE-4-12 and SMoE-4-4-4-4. The SMoE-4-12 configuration consistently demonstrates superior performance, achieving a larger margin compared to our best baselines, EOM and CMR. The SMoE-4-12 outperforms our naive MoE model and Switch Transformer by +1.01 and +1.63 BLEU, significantly outperforming the gains achieved by EOM (+0.11 and +0.73) and CMR (+0.14 and 0.76).

Overall, SMoE outperforms all baselines with the same or a fewer number of parameters.

### 4.6 Computational Cost

As a token may pass through multiple strata in any given SMoE layer, the average computational cost is higher than other MoE models. In the last column of Tables 1 and 2, we record FLOPs per token for all models, and Table 3 shares the

| Methods | eng→xxx | | | | xxx→eng | | | | Avg. |
|---|---|---|---|---|---|---|---|---|---|
| | all | high | medium | low | all | high | medium | low | |
| Dense Model (Vaswani et al., 2017) | 27.37 | 23.89 | 31.17 | 29.76 | 30.60 | 29.40 | 31.85 | 31.49 | 28.99 |
| Vanilla MoE (Lepikhin et al., 2021) | 30.34 | 26.16 | 34.78 | 33.38 | 32.38 | 31.20 | 33.67 | 33.21 | 31.36 |
| Switch Transformer (Fedus et al., 2021) | 30.03 | 25.76 | 34.46 | 33.27 | 31.44 | 30.58 | 33.00 | 31.18 | 30.74 |
| MoE+EOM (NLLB Team et al., 2022) | 30.48 | 26.24 | 34.76 | 33.86 | 32.45 | 31.23 | 33.69 | 33.44 | 31.47 |
| MoE+CMR (NLLB Team et al., 2022) | 30.56 | 26.32 | 34.98 | 33.77 | 32.43 | 31.33 | 33.69 | 33.11 | 31.50 |
| SMoE-4-12 | **32.15** | **27.99** | **36.67** | 35.06 | 32.58 | **32.42** | 33.76 | 31.37 | **32.37** |
| SMoE-4-4-4-4 | 31.84 | 27.31 | 36.52 | **35.33** | **32.71** | 31.87 | 33.80 | 33.04 | 32.28 |

Table 3: Overall BLEU results on the OPUS-100 dataset. The best values are bold and the second-best values are underlined. The number of experts is 16. We consider the two best settings in M15 dataset, SMoE-4-12 and SMoE-4-4-4-4. Both of them substantially outperform all baselines. The number of parameters and FLOPs/tok for MoE models are the same as Table 2.

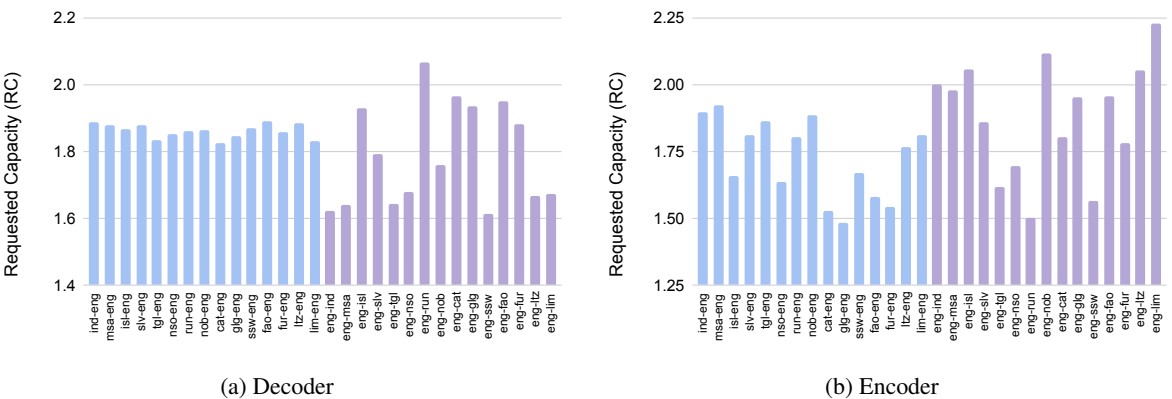

(a) Decoder           (b) Encoder

Figure 2: Average requested capacity (RC) of all tokens in each translation direction. The blue bars are for xxx→eng directions and the purple bars are for eng→xxx directions. Directions in each subset are sorted from high-resource to low-resource. On the decoder side, the average RC of eng tokens is similar regardless of the source language, but averaged RC has a large variance if the target language is different. On the encoder side, RC is always different even though the source language is the same.

same information with Table 2.[4] Although SMoE requires more FLOPs per token, the additional cost is only a marginal increase over vanilla MoE models. For example, in M15 and OPUS-100, our best setting SMoE-4-12 merely uses 8% more FLOPs/tok than other top-2-gating MoE models, but significantly outperforms all of them.

## 5 Analysis

The advantage of SMoE is that it can assign dynamic capacity to different tokens, with some tokens passing through only one expert and others passing through multiple experts. Here, we define the **Requested Capacity (RC)** as the average number of experts that a token need to pass in one SMoE block. RC of a token is dictated by how the SMoE gates route it throughout the different strata. To understand what may affect the RC of tokens,

we examine three potential influencing factors: the language of the input token, the frequency of the token, and the depth of the SMoE block. All analysis is conducted using the SMoE-4-4-4-4 model trained on the M15 dataset.

### 5.1 The Language of The Input Token

Here, we investigate whether different languages have different RCs. We begin with collecting the average RC in all translation directions for all tokens in the training and development (Flores-200 dev) sets. We investigate SMoE blocks in the encoder and decoder separately as they process different tokens (source tokens for the encoder and target tokens for the decoder). The average RC is then averaged across all SMoE blocks in either encoder or decoder.

Figure 2a shows the average RC in the decoder for each translation direction in M15. When translating into English (xxx→eng, blue bars), we

---
[4]FLOPs are calculated for the forward pass as done in Kaplan et al. (2020).

observe that the target English tokens have similar RC in the decoder side (≈1.85 experts) irrespective of the source language. When translating from English (eng→xxx), the RC varies a lot with respect to the target language.

Unlike in the decoder where only the target language matters, Figure 2b shows variability in RC with respect to both source and target languages i.e, not only the language of the tokens themselves (source), but also the target language we will be translating into once we move to the decoder. We hypothesize the special symbol at the beginning of the source sequence (<2xxx>) can affect the capacity assignment. In conclusion, capacity assignment is sensitive to the target language in the decoder and to the translation direction in the encoder.

## 5.2 Token Frequency

As in the previous section, we record the average RC for all tokens in training and development data, and in all translation directions. To avoid looking at all 32K tokens in our vocabulary, we select the top-25 tokens with the highest RC in each SMoE block and in each translation direction, totaling 4500 tokens.[5] We similarly collect the bottom-25 tokens with the lowest RC. After removing tokens repeatedly selected by different directions or by different SMoE blocks, we end up with 2580 unique high-RC tokens and 3208 unique low-RC tokens. We draw in Figure 3 a violin plot to show the distribution of tokens in these two groups in terms of their frequency in the training data. We rank the frequencies on the y-axis so that a lower rank means more frequent tokens, e.g., rank 0 corresponds to the most frequent token in our training data. The results show that there is no strong correlation between frequency and RC for tokens with the highest RC. On the other end of the spectrum, tokens with the lowest RC tend to be high-frequency tokens, as indicated by the right violin plot being wider at the bottom part (rank < 10k). Many of these high-frequency tokens are basic subword units (like _li, _el, _pa) or punctuation marks. One can interpret RC as a metric for 'difficulty in processing a given token'. The model was overly exposed to these frequent tokens, and as such, does not require a lot of capacity to process them.

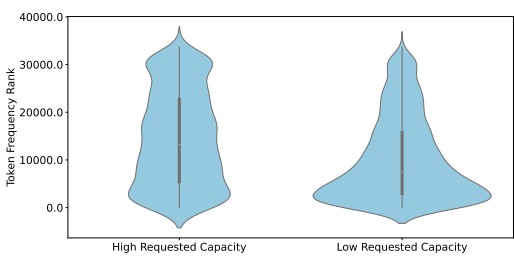

Figure 3: Violin plots of the token frequency in high-RC (left) and low-RC (right) tokens. Unlike high-RC tokens, low-RC tokens tend to be highly frequent ones.

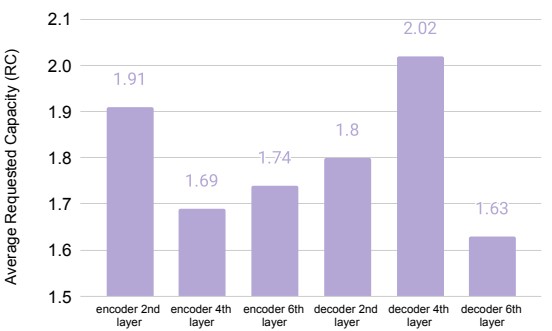

Figure 4: Average RC of SMoE blocks in different locations of the architecture.

## 5.3 Location of The SMoE Block

We analyze in this section the average RC in relation to the location of the SMoE block in the transformer architecture. As depicted in Figure 4, RC varies depending on the location of the SMoE block. Early encoder layers (encoder 2nd layer is the first SMoE block in the model) request more capacity than the subsequent encoder SMoE blocks. We hypothesize that this first block takes on the task of mapping tokens coming from different languages and different scripts to a shared space.

## 6 Conclusion

This work presents Stratified Mixture of Experts (SMoE) models, a novel design for MoEs that is capable of dynamically assigning capacity to input tokens. Through experimental evaluation at three scales (M4, M15, and OPUS-100), we have demonstrated that the proposed SMoE model surpasses the performance of many current state-of-the-art MoE methods. This proves that dynamically assigning capacity to tokens in MoE models is a viable solution to address the MoE's parameter inefficiency. Additionally, we conduct a thorough analysis to investigate the factors that

---

[5]$25 \times 6$ (#SMoE blocks) $\times 30$ (#directions) = 4500.

influence dynamic capacity assignment, including the language of the tokens and the location of the SMoE block within the model architecture.

## Limitations

Stratified Mixture of Experts (SMoE) aims to improve the performance of Mixture of Experts (MoE) models by assigning dynamic capacity to different tokens. While SMoE has demonstrated performance improvements over many state-of-the-art baselines, it also comes with an additional computational cost compared to traditional MoE models. However, the cost is small and the benefits of SMoE in terms of improved performance often outweigh this added computational cost, especially in tasks where performance is critical. For example, in OPUS-100, with 8% FLOPs/tok, SMoE-4-12 achives +1.01 BLEU compared with traditional MoE (Lepikhin et al., 2021).

## Acknowledgements

We thank anonymous reviewers for their insightful feedback. We also extend our gratitude to Steven Tan and Yunmo Chen for their valuable suggestions.

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

## A   M15 Information

We show the detailed information of the M15 dataset in Table 4.

## B   Additional Training Details

We employ the transformer$_{base}$ model (with an FFN dimension of 2048 and an embedding dimension of 512) for the M4 dataset, and the transformer$_{big}$ model (with an FFN dimension of 4096 and an embedding dimension of 1024) for M15 and OPUS-100 dataset. The maximum learning rate is 0.0008 for M4 and M15, and 0.0005 for the OPUS-100 dataset. The optimizer is Adam (Kingma and Ba, 2014) with `inverse_sqrt` learning rate scheduler and weight decay of 0. The total number of training steps is 100K with 8K warm-up steps. The batch size is 13K tokens for M4 and M15, and 65K tokens for OPUS-100.

| Language | Language code | Parallel Data Size | Resource Level | Language family |
|---|---|---|---|---|
| Northern Sotho | nso | 526 K | Low | Central Narrow Bantu |
| Rundi | run | 454 K | Low | Central Narrow Bantu |
| Swati | ssw | 94 K | Very Low | Central Narrow Bantu |
| Indonesian | ind | 6.5 M | High | Malayo-Polynesian |
| Malay | msa | 1 M | High | Malayo-Polynesian |
| Tagalog | tgl | 1 M | High | Malayo-Polinesian |
| Bokmål (Norwegian) | nob | 238 K | Low | North Germanic |
| Icelandic | isl | 1 M | High | North Germanic |
| Faroese | fao | 4 K | Very Low | North Germanic |
| Slovene | slv | 15 M | High | Southwestern Slavic |
| Luxembourgish | ltz | 8 K | Very Low | Western Germanic |
| Limburgish | lim | 5 K | Very Low | Western Germanic |
| Catalan | cat | 634 K | Low | Western Romance |
| Galician | glg | 195 K | Low | Western Romance |
| Friulian | fur | 6 K | Very Low | Western Romance |

Table 4: Information about the 15 languages in the M15 dataset.