# OpenReview forum: "Towards Being Parameter-Efficient: A Stratified Sparsely Activated Transformer with Dynamic Capacity"
_EMNLP/2023/Conference — EMNLP 2023 Findings_

### Official Review · Reviewer_zL3X · 2023-07-23

**Soundness:** 3

**Excitement:**

3: Ambivalent: It has merits (e.g., it reports state-of-the-art results, the idea is nice), but there are key weaknesses (e.g., it describes incremental work), and it can significantly benefit from another round of revision. However, I won't object to accepting it if my co-reviewers champion it.

**Paper Topic And Main Contributions:**

This paper proposes a new architecture for mixture-of-experts (MoE) models called Stratified Mixture of Experts (SMoE). The key ideas and contributions are:

1. MoE models suffer from parameter inefficiency, where adding more experts does not improve performance proportionally. The authors hypothesize this is due to all experts having equal capacity, while different tokens may require different amounts of capacity.

2. SMoE assigns dynamic capacity to tokens by having multiple strata of experts. Tokens can pass through more experts and get more capacity if routed across multiple strata. This allows more efficient use of parameters.

3. SMoE is evaluated on multilingual machine translation across 3 datasets with 4, 15 and 94 languages. It substantially outperforms strong MoE baselines like Switch Transformer, MoE+EOM and MoE+CMR with the same or fewer parameters.

4. Analysis shows the capacity assignment in SMoE depends on factors like target language, source language, token frequency and location of the SMoE block.

**Questions For The Authors:**

Could you give me explanation or external information for my reasons to reject?

**Reasons To Accept:**

1. It tackles an important problem in improving the parameter efficiency of mixture-of-experts (MoE) models. Recent work has shown that MoE models tend to be parameter inefficient, so addressing this issue could lead to more efficient large-scale models.

2. The proposed Stratified Mixture of Experts (SMoE) model introduces a novel stratified architecture that allows dynamic capacity assignment to tokens. This concept of adaptive capacity seems promising for improving model performance.
The paper demonstrates strong empirical results, substantially outperforming several state-of-the-art MoE models on multilingual machine translation benchmarks while using the same or fewer parameters. The gains are particularly pronounced on the largest dataset with 94 languages.

3. The analysis provides useful insights into what factors affect the capacity assignment, shedding light on how the model adapts capacity based on token language, frequency, and position in the architecture.

**Reasons To Reject:**

1. You propose that this parameter inefficiency arises from all experts having equal capacity, and then introduce SMoE as a solution, assigning dynamic capacity to different tokens. However, the resulting performance improvement of the architecture is marginal (MoE+CMR 40.35 vs. SMoE-2-2-2-2 40.78). Similar observations are evident in other experiments. Consequently, this evidence is insufficient to validate your hypothesis.

2. The paper focuses solely on machine translation tasks. The benefits of the proposed SMoE model for other NLP tasks are unclear. Testing on a broader set of tasks would strengthen the paper.

3. The paper focuses only on token-level of MoE. Testing on task-level and sequence-level would strengthen the paper.

4. The comparison is only to other Mixture-of-Experts models. Comparing to state-of-the-art non-MoE models on machine translation could better highlight if the improvements are coming from the stratified architecture specifically or just having more parameters/capacity in general.

5. The increased computational cost of SMoE compared to standard MoE is not thoroughly analyzed. The trade-off between improved performance and higher FLOPs should be clarified.

6. The SMoE model requires carefully tuning the number of strata and experts per strata for good performance. The paper does not provide much insight into how to set these hyperparameters.

**Reproducibility:**

4: Could mostly reproduce the results, but there may be some variation because of sample variance or minor variations in their interpretation of the protocol or method.

**Reviewer Confidence:**

4: Quite sure. I tried to check the important points carefully. It's unlikely, though conceivable, that I missed something that should affect my ratings.

---

> ### Author Rebuttal · Authors · 2023-08-29
>
> We genuinely appreciate the valuable feedback provided by the reviewer and have addressed them in a point-by-point manner below. We are more than willing to engage in further discussions with the reviewers should any follow-up questions arise.
>
> ### **Regarding your concern 1**
> >However, the resulting performance improvement of the architecture is marginal (MoE+CMR 40.35 vs. SMoE-2-2-2-2 40.78).
>
> Thank you for the insightful feedback. It's imperative to underscore that while the absolute difference may appear marginal in smaller datasets, these results are averaged over 8, 30, and 190 directions. The intricacies of the dataset can potentially reduce the improvements for all MoE methodologies. Taking the example from the reviewer, indeed, SMoE outperforms CMR on M4 by +0.43 BLEU. However, SMoE-2-2-2-2 outperforms vanilla MoE by +0.74 BLEU but CMR only achieves  +0.31 BLEU improvement.
>
> Furthermore, our methods consistently outperform all strong baselines across all three MMT datasets (M4, M15, and OPU-100), encapsulating varying data size distributions and linguistic diversity. Notably, in OPUS-100, our SMoE registers a commendable +1.01 BLEU advantage over vanilla MOE, whereas CMR, despite demanding more parameters and FLOPs/tok, provides just a +0.14 BLEU increment.
>
> ### **Regarding your concern 2**
> >The paper focuses solely on machine translation tasks.
>
> Thank you for your insightful comments. We wish to emphasize that multilingual machine translation is particularly apt for evaluating the efficacy of MoE models. This is because it mandates MoE models to handle tokens from diverse languages and resources, in order to mitigate negative language interference. Additionally, the parameter inefficiency is also usually observed under task that needs a large amount (millions) of datasets, where only machine translation data has this scale
>
> As highlighted in line 102, our goal centers on the multilingual machine translation task. Nonetheless, we concur with your viewpoint that expanding the scope to encompass more tasks could further enrich and fortify our paper.
>
> ### **Regarding your concern 3**
> >The paper focuses only on token-level of MoE. Testing on task-level and sequence-level would strengthen the paper.
>
> Thank you for your insightful feedback. Our exploration of SMoE models stems from the hypothesis that tokens from distinct languages may require varying capacities. Consequently, we begin with token-level MoE, upon which many strong MoE systems for MMT are built. This approach also benefits from expert parallelism due to the (almost) even distribution of tokens across experts by using load balance loss. Nonetheless, we concur with your suggestion that incorporating task-level or sequence-level perspectives would be beneficial.
>
> ### **Regarding your concern 4**
> >Comparing to state-of-the-art non-MoE models on machine translation could better highlight if the improvements are coming from the stratified architecture specifically or just having more parameters/capacity in general.
>
> Thank you for your constructive feedback. We would like to first highlight that the improvements have been proved coming from the stratified architecture because our SMoE is compared with other MoE models under the same number of parameters.
>
> Secondly, it is hard to define what is the sota non-MMT model now because of the difference in focused languages, model size, and settings (like different vocabulary). Moreover, it has been widely proved that MoE model outperforms the non-MoE models under the same FLOPs/tok. For example, NLLB-EOM and CMR is better than its dense model which is trained on an extensive dataset, utilizing back translation and curriculum learning.
>
> However, we value and follow the suggestion of the reviewer and conduct a non-MoE method on OPUS-100. The method we pick is CLSR [1], which is not only widely recognized as a strong MMT model but also shares a similar spirit/motivation with our study in the context of multilingual neural machine translation. It introduces a gating mechanism that controls whether a given token should be processed by a language-specific parameter or shared parameter (but their capacity for tokens is fixed).  As illustrated below, our approach markedly surpasses CLSR in average performance:
>
> | Methods   | en-xx all | en-xx high | en-xx med | en-xx low | xx-en all | xx-en high | xx-en med | xx-en low | avg.      |
> |-----------|:---------:|:----------:|:---------:|:---------:|-----------|------------|-----------|-----------|-----------|
> | CLSR      | 27.39     | 23.91      | 31.17     | 29.77     | 32.43     | 30.73      | **33.85** | **34.20** | 29.91     |
> | SMoE-4-12 | **32.15** | **27.99**  | **36.67** | **35.06** | **32.58** | **32.42**  | 33.76     | 31.37     | **32.37** |
>
> Reference:
> [1] Zhang B, Bapna A, Sennrich R, Firat O. Share or not? learning to schedule language-specific capacity for multilingual translation. International Conference on Learning Representations 2020 Oct 2.
>
> ### **Regarding your concern 5 and concern 6**
> >The increased computational cost of SMoE compared to standard MoE is not thoroughly analyzed. The trade-off between improved performance and higher FLOPs should be clarified.
>
> Thanks for your insightful feedback!  To understand the optimal configuration of SMoE, we have a comprehensive hyperparameter search in Table 2 including varying the depths and number of experts, and we report all their detailed performance and the FLOPs/tok they cost. While no straightforward correlation emerged between improved performance and higher FLOPs/tok, our findings did spotlight SMoE-4,12 as the most efficient design, incurring only an 8% overhead in FLOPs/tok.
>
> >The SMoE model requires carefully tuning the number of strata and experts per strata for good performance. The paper does not provide much insight into how to set these hyperparameters.
>
> We identified a pattern where a shallow SMoE with 1/3 experts in lower strata and 2/3 experts in higher strata performs best. This trend held true across experiments on both M15 and OPUS-100. Based on these observations, we would recommend following a similar strategy of stratified layers on other benchmarks. We will make sure to mention this in our conclusion.

---

### Official Review · Reviewer_3H55 · 2023-08-03

**Soundness:** 4

**Excitement:**

4: Strong: This paper deepens the understanding of some phenomenon or lowers the barriers to an existing research direction.

**Missing References:**

Another Parameter Efficient MoE:
https://arxiv.org/abs/2107.11817

**Paper Topic And Main Contributions:**

This paper devotes to making MoE-based transformers more parameter efficient. To this end, the authors propose a Stratified Mixture of Experts (SMoE). That is, authors use a stratified MoE structure that enables different tokens to have different capacities. The SMoE model outperforms many popular MoE baselines.

**Reasons To Accept:**

1) Stratified MoE structure is very novel and interesting. Different tokens can also have an adaptive computation budget in this case, which may make many impossible tasks possible. (See [1])
2) The paper is well-written and easy to read.
3) Section 5 is great. It conveys a more comprehensive understanding about how the model works.

[1] https://arxiv.org/abs/1807.03819

**Reasons To Reject:**

1) How compatible is the SMoE with expert parallelism? It seems that the stratified structure requires more carefully implemented expert parallelism. Or authors actually ignored the expert parallelism in this paper?
2) The baselines are indeed popular but it is a bit overclaim to say these MoE models are SoTA. We have many better models like Brainformer and Expert-choice MoE. It is okay to compare these popular models since some of these advanced models are not open-sourced, but please do not call them SoTA.


Minor: Fig 1b and Fig 1c are almost the same. It is okay to keep 1b only.

**Reproducibility:**

3: Could reproduce the results with some difficulty. The settings of parameters are underspecified or subjectively determined; the training/evaluation data are not widely available.

**Reviewer Confidence:**

5: Positive that my evaluation is correct. I read the paper very carefully and I am very familiar with related work.

---

> ### Author Rebuttal · Authors · 2023-08-29
>
> We genuinely appreciate the valuable feedback provided by the reviewer and have addressed them in a point-by-point manner below. We are more than willing to engage in further discussions with the reviewers should any follow-up questions arise.
>
> ### **Regarding your concern about expert parallelism**
> >How compatible is the SMoE with expert parallelism?
>
> Thanks for your detailed question! We always consider expert parallelism and use ALLTOALL communication, following the algorithm in Gshard [1]. We evenly distribute experts to our GPUs. If we have 16 experts and 4 GPUs, we group every 4 experts into one GPU regardless of the stratum of where they are located. We will clarify this in the paper.
>
> Reference:
> [1] Lepikhin D, Lee H, Xu Y, Chen D, Firat O, Huang Y, Krikun M, Shazeer N, Chen Z. Gshard: Scaling giant models with conditional computation and automatic sharding. arXiv preprint arXiv:2006.16668. 2020 Jun 30.
>
> ### **Regarding your concern about overclaiming Sota**
> >The baselines are indeed popular but it is a bit overclaim to say these MoE models are SoTA
>
> Thanks for your comments! We would like to clarify that our MoE model is designed to focus on the multilingual machine translation task. Consequently, we selected models that align with our goal (given that expert-choice MoE isn't primarily designed for MMT). Notably, NLLB-EOM/CMR stands out as one of the most outstanding MoE models tailored for MMT. This is the main reason for adopting them as baselines — making the baselines more relevant and meaningful. We acknowledge that Brainformer, being a concurrent work, wasn't available when our study was finalized. Nevertheless, we concur with the reviewer's perspective on the value of incorporating a broader range of baselines, and we appreciate the suggestion to gauge SMoE's standing amidst other robust MoE models.

---

### Official Review · Reviewer_T5W3 · 2023-08-03

**Soundness:** 3
**Typos Grammar Style And Presentation Improvements:** line 263 'extracte' --> 'extracted'.

**Excitement:**

3: Ambivalent: It has merits (e.g., it reports state-of-the-art results, the idea is nice), but there are key weaknesses (e.g., it describes incremental work), and it can significantly benefit from another round of revision. However, I won't object to accepting it if my co-reviewers champion it.

**Missing References:**

I suggest adding references (paper and resources) for the training data NLLB into the paper.

**Paper Topic And Main Contributions:**

This work proposes a new model called Stratified Mixture of Experts (SMoE) to address the issue of parameter inefficiency in Mixture-of-experts (MoE) models. The authors hypothesize this issue is a result of all experts having equal model capacity, which cannot provide adaptive complexity requirements of different tokens or tasks. They propose to build an MoE system with a stratified structure that can assign dynamic capacity to different tokens. They conducted experiments on two multilingual machine translation benchmarks, where it outperforms multiple MoE models with on-par or fewer model parameters.

**Reasons To Accept:**

1. The paper develops a Stratified Mixture of Experts (SMoE) to address the parameter inefficiency in Mixture-of-experts (MoE) models, which is interesting and reasonable.

2. The SMoE outperforms multiple MoE models on multiple machine translation tasks impressively.

**Reasons To Reject:**

1. The technical contribution is incremental. The main novelty comes from restructuring the MoE system to dynamically assign model capacities for different inputs.

2. The experiment is limited to machine translation tasks and some efficient MoE baselines are not compared [1].

3. The comparison between SMoE and other Sota methods for MT is missing, hence it is not clear whether it is necessary to have multiple experts.

[1] Parameter-Efficient Mixture-of-Experts Architecture for Pre-trained Language Models. Gao et al. COLING22.

**Reproducibility:**

4: Could mostly reproduce the results, but there may be some variation because of sample variance or minor variations in their interpretation of the protocol or method.

**Reviewer Confidence:**

3: Pretty sure, but there's a chance I missed something. Although I have a good feel for this area in general, I did not carefully check the paper's details, e.g., the math, experimental design, or novelty.

---

> ### Author Rebuttal · Authors · 2023-08-29
>
> We genuinely appreciate the valuable feedback provided by the reviewer and have addressed them in a point-by-point manner below. We are more than willing to engage in further discussions with the reviewers should any follow-up questions arise.
>
> ### **Regarding your concern about incremental contribution**
> >The technical contribution is incremental. The main novelty comes from restructuring the MoE system to dynamically assign model capacities for different inputs
>
> Thanks for your comments! Regarding your concern about the incremental contribution, we are not claiming a fundamental change to an underlying approach. Moreover, we also provided complete experiments and strong baselines to support the effectiveness of our methods. Indeed, our hope is that the relative ease in understanding and reproducing our proposals is why they are most likely to be picked up by the community. We would be happy if the common reaction by the reader was, 'Hunh, that seems easy!' we would say that a key point of this article is the relative straightforwardness of the proposed.
>
> ### **Regarding your concern about experiments**
> >The experiment is limited to machine translation tasks
>
> Thank you for your insightful comments. We wish to emphasize that multilingual machine translation is particularly apt for evaluating the efficacy of MoE models. This is because it mandates MoE models to handle tokens from diverse languages and resources, in order to mitigate negative language interference. Additionally, the parameter inefficiency is also usually observed under task that needs a large amount (millions) of datasets, where only machine translation data has this scale. As highlighted in line 102, our goal centers on the multilingual machine translation task, and to deepen our study, we conduct comprehensive experiments from 3 different types of MMT datasets (ranging from 4 to 95 languages). Nonetheless, we concur with your viewpoint that expanding the scope to encompass more tasks could further enrich and fortify our paper.
>
> >and some efficient MoE baselines are not compared [1].
>
> We'd like to clarify that the methodology presented in [1] follows a distinct trajectory in studying MoE models. Their approach leverages pre-trained architectures like T5 or GPT2, whereas our model is designed and trained from scratch. Crucially, when selecting our baselines, we opted for MoE model architectures that shared our primary objective of enhancing multilingual machine translation. This ensures that our comparisons are both robust and meaningful. Furthermore, our experiments on MMT are comprehensive, spanning from 4 to 95 languages, which provides a holistic perspective on language distribution and diversity within the task.
>
> Reference:
> [1] Parameter-Efficient Mixture-of-Experts Architecture for Pre-trained Language Models. Gao et al. COLING22.
>
> ### **Regarding your concern about comparing with other SOTA non-MoE  MMT methods**
> >The comparison between SMoE and other Sota methods for MT is missing, hence it is not clear whether it is necessary to have multiple experts.
>
> Thank you for your constructive feedback! It is hard to define what is the Sota non-MMT model because of the difference in focused languages, model size, and settings (like different vocabulary). Moreover, it has been widely proved that MoE model outperforms the non-MoE models under the same FLOPs/tok. For example, NLLB-EOM and CMR [1] are better than their dense model which is trained on an extensive dataset, utilizing back translation and curriculum learning.
>
> However, we value and follow the suggestion of the reviewer and conduct a non-MoE method on OPUS-100. The additional baseline we pick is CLSR [2], which is not only widely recognized as a strong MMT model but also shares a similar spirit/motivation with our study in the context of multilingual neural machine translation. It introduces a gating mechanism that controls whether a given token should be processed by a language-specific parameter or shared parameter (but their capacity for tokens is fixed).  As illustrated below, our approach markedly surpasses CLSR in average performance:
>
> | Methods   | en-xx all | en-xx high | en-xx med | en-xx low | xx-en all | xx-en high | xx-en med | xx-en low | avg.      |
> |-----------|:---------:|:----------:|:---------:|:---------:|-----------|------------|-----------|-----------|-----------|
> | CLSR      | 27.39     | 23.91      | 31.17     | 29.77     | 32.43     | 30.73      | **33.85** | **34.20** | 29.91     |
> | SMoE-4-12 | **32.15** | **27.99**  | **36.67** | **35.06** | **32.58** | **32.42**  | 33.76     | 31.37     | **32.37** |
>
> Reference:
> [1] NLLB Team, Costa-jussà MR, Cross J, Çelebi O, Elbayad M, Heafield K, Heffernan K, Kalbassi E, Lam J, Licht D, Maillard J, Sun A. No language left behind: Scaling human-centered machine translation. arXiv preprint arXiv:2207.04672. 2022 Jul 11.
> [2] Zhang B, Bapna A, Sennrich R, Firat O. Share or not? learning to schedule language-specific capacity for multilingual translation. International Conference on Learning Representations 2020 Oct 2.

---

### Meta-Review · Area_Chair_qu6W · 2023-09-25

**Recommendation:** 3

**Metareview:**

This paper proposes a dynamic capacity approach for MoE models to mitigate parameter inefficiency. The reviews expressed concerns about (i) the necessity of applying the approach to tasks beyond MT, (ii) the marginal improvements in MT, (iii) the time requirements and computational efficiency of the approach. The rebuttal has (partly) addressed the concerns. Stronger results on MT and applying the approach n tasks beyond MT certainly strengthen the paper in future.

---

### Decision · Program_Chairs · 2023-10-07

**Decision:**

Accept-Findings

**Comment:**

This paper proposes a dynamic capacity approach for MoE models to mitigate parameter inefficiency. The reviews expressed concerns about (i) the necessity of applying the approach to tasks beyond MT, (ii) the marginal improvements in MT, (iii) the time requirements and computational efficiency of the approach. The rebuttal has (partly) addressed the concerns. Stronger results on MT and applying the approach n tasks beyond MT certainly strengthen the paper in future.